# Group Excess Risk Bound of Overparameterized Linear Regression with Constant-Stepsize SGD

**Arjun Subramonian**
Meta AI, UCLA

**Levent Sagun**
Meta AI

**Kai-Wei Chang**
UCLA

**Yizhou Sun**
UCLA

## Abstract

It has been observed that machine learning models trained using stochastic gradient descent (SGD) exhibit poor generalization to certain groups within and outside the population from which training instances are sampled. This has serious ramifications for the fairness, privacy, robustness, and out-of-distribution (OOD) generalization of machine learning. Hence, we theoretically characterize the inherent generalization of SGD-learned overparameterized linear regression to intra- and extra-population groups. We do this by proving an excess risk bound for an arbitrary group in terms of the full eigenspectra of the data covariance matrices of the group and population. We additionally provide a novel interpretation of the bound in terms of how the group and population data distributions differ and the group effective dimension of SGD, as well as connect these factors to real-world challenges in practicing trustworthy machine learning. We further empirically study our bound on simulated data.

## 1 Introduction

Much recent work has sought to better understand the inductive biases of stochastic gradient descent (SGD), such as benign overfitting and implicit regularization in overparameterized settings [1, 2, 3]. However, this line of literature has overwhelmingly focused on bounding the excess risk of an SGD-learned model over the entire population $\mathcal{P}$ from which training instances are sampled, and has not investigated how SGD (e.g., its assumption that training data are IID) yields poor model generalization to intra-population groups $\mathcal{G}_{\text{intra}}$ (i.e., subsets of the population) and extra-population groups $\mathcal{G}_{\text{extra}}$ (i.e., instances that fall outside the population). We illustrate these concepts in Figure 1.

Establishing the theory behind this phenomenon is critical, as it provides provable guarantees about the trustworthiness (e.g., fairness, privacy, robustness, and out-of-distribution generalization) of SGD-learned models. As an example, let's consider an automated candidate screening system that a company trains on the qualifications data (e.g., number of years previously worked, relevant skills) of a sample from the population $\mathcal{P}$ of past job applicants [4]. In the context of fairness, many works have observed that for models trained using SGD, group-imbalanced data distributions translate to generalization disparities [5, 6, 7, 8]. Hence, the candidate screening system may generalize poorly for minoritized groups $\mathcal{G}_{\text{intra}}$ (i.e., not satisfy equalized odds [9]), which can yield hiring discrimination. This phenomenon also has implications for privacy, as adversaries, against the desire of a job applicant, can infer whether the applicant's data were used to train the candidate screening system, based on the system's loss on the applicant [10]. When considering robustness, we may be interested in how well the candidate screening system generalizes to a target group $\mathcal{G}_{\text{intra}}$ of applicants when $\mathcal{P}$ is noisy or corrupted [11, 12]. Finally, in the context of out-of-distribution generalization, models deployed in the real world often have to deal with data distributions that differ from the training distribution [13]; for instance, the candidate screening system, if trained prior to a recession, may generalize poorly to stellar job applicants $\mathcal{G}_{\text{extra}}$ who were laid off during the recession for reasons beyond their control. Therefore, it is paramount to understand how SGD-learned models

2022 Trustworthy and Socially Responsible Machine Learning (TSRML 2022) co-located with NeurIPS 2022.

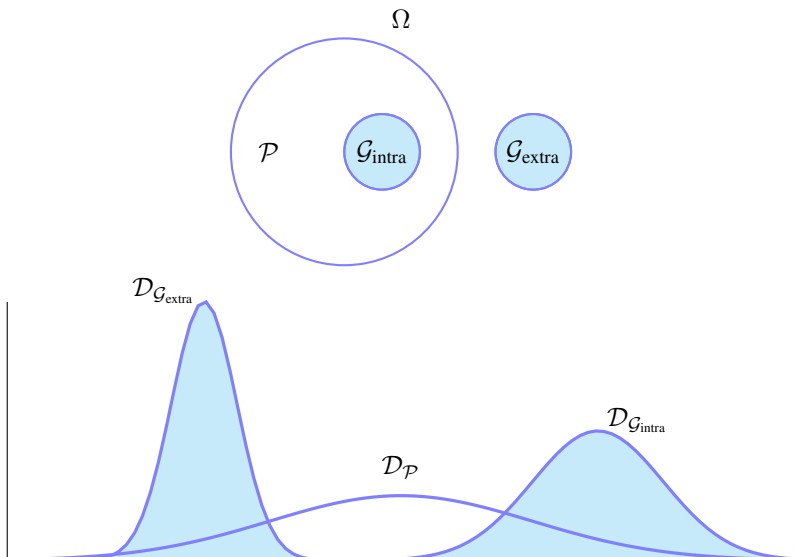

Figure 1: Euler diagram of a population $\mathcal{P}$, an intra-population group $\mathcal{G}_{\text{intra}}$, and an extra-population group $\mathcal{G}_{\text{extra}}$, as well as a visual depiction of their respective possible data distributions $\mathcal{D}_{\mathcal{P}}, \mathcal{D}_{\mathcal{G}_{\text{intra}}}, \mathcal{D}_{\mathcal{G}_{\text{extra}}}$ (which are noticeably distinct).

generalize to extra-population groups [14], especially in terms of how the properties of the data distributions for these groups differ from those of the training distribution.

Towards bolstering the theoretical foundations of trustworthy machine learning, we characterize the inherent generalization of constant-stepsize SGD (with iterate averaging) to groups within and outside the population from which training instances are sampled, for an arguably simple setting: overparameterized linear regression. We extend the analysis of [1] to prove an excess risk bound for an arbitrary group which can be decomposed into bias and variance factors that depend on the full eigenspectra of the data covariance matrices of the group and population. We then re-express the excess risk bound in terms of: 1) how the group and population data distributions diverge, 2) how the distributions' feature variances differ, and 3) the group effective dimension of SGD. We connect these three components to real-world challenges in practicing trustworthy machine learning, such as limited features and sample size disparities for minoritized groups. Finally, we empirically study our bound on simulated data. As a whole, we set the stage for future research to extend our results to deep neural networks and models trained with other variants of gradient descent (e.g., minibatch GD, SGD with learning rate scheduling).

## 2 Related work

**Imbalanced learning** The tendency of machine learning models to overpredict the majority class in the presence of class-imbalanced training samples [6, 15, 7, 16, 8, 17] and underperform for minoritized groups [18, 19] has been extensively empirically and theoretically studied. Some papers have theoretically investigated worst-case group generalization in the overparameterized regime [20, 21]. However, these works have not examined how SGD in particular (e.g., its assumption that training data are IID) causes poor generalization for a minoritized group, even in the arguably simple case of overparameterized linear regression. We do so in terms of the data covariance matrices of the group and population, rather than representation dimension or information, which affords greater interpretability.

**Inductive biases of SGD** Theoretically analyzing the inductive biases of stochastic gradient descent (e.g., implicit regularization, benign overfitting), especially in the overparameterized regime, is a nascent area of research [2, 1, 3, 22, 23, 24] and strengthens our understanding of how deep learning works. In this paper, we make novel contributions to learning theory by analyzing constant-stepsize SGD with iterate averaging when training instances are sampled from a different distribution than the

evaluation distribution. We do so by extending the analysis of [1], which only analyzes the excess risk of SGD-learned linear regression over the *entire* population from which training instances are sampled, and not over a particular group.

**Fair machine learning**  Numerous works in fair machine learning have explored the implications of generalization disparities among groups (known as equalized odds [9]) for model-induced harms faced by minoritized groups [25, 5]. For example, in the case of automated loan approval, if white men enjoy better model generalization, their loan applications will be less likely to be incorrectly rejected compared to women and people of color. Prior research has also theoretically and empirically studied worst-case group generalization in the context of fairness without demographics and distributionally robust optimization [26, 27, 28]. In this work, motivated by the equalized odds framing of fairness, we prove a group excess risk bound for overparameterized linear regression and contextualize the bound in terms of real-world challenges in practicing fair machine learning.

## 3   Problem setup

We consider the linear regression problem $\min_w L_{\mathcal{D}}(w)$, where $L_{\mathcal{D}}(w) = \frac{1}{2}\mathbb{E}_{(x,y)\sim\mathcal{D}}\left[(y - \langle w, x\rangle)^2\right]$. In this equation, $x \in \mathcal{H}$ is the feature vector (where $\dim(\mathcal{H})$ can be but need not be $\infty$ to model the overparameterized regime, wherein $d >> N$), $y \in \mathbb{R}$ is the response, $w \in \mathcal{H}$ is the weight vector to be optimized, and $\mathcal{D}$ is the arbitrary population distribution over $x$ and $y$. Furthermore, suppose an arbitrary group $m$ (within or outside the population) has the arbitrary distribution $\mathcal{D}_m$ over $x$ and $y$. Now, assume the unique optimal parameters $w_m^*$ for group $m$ satisfy the first-order optimality condition $\nabla L_{\mathcal{D}_m}(w_m^*) = \mathbb{E}_{(x,y)\sim\mathcal{D}_m}\left[(y - \langle w_m^*, x\rangle) x\right] = \mathbb{E}_{(x,y)\sim\mathcal{D}_m}\left[\xi x\right] = 0$.

In this paper, we consider constant stepsize SGD with iterate averaging; at each iteration $t$, a training instance $(x_t, y_t) \sim \mathcal{D}$ is independently observed and the weight is updated as follows:

$$w_t := w_{t-1} - \gamma \left(\langle w_{t-1}, x_t\rangle - y_t\right) x_t, t = 1, \ldots, N$$

where $\gamma > 0$ is a constant stepsize, $N$ is the number of samples observed, and the weights are initialized as $w_0 \in \mathcal{H}$. Following [1], the final output is the average of the iterates $\bar{w}_N := \frac{1}{N}\sum_{t=0}^{N-1} w_t$.

## 4   Main result

We now introduce relevant notation and our assumptions (which are similar to those in [1]), as well as state our main result.

**Assumption 1 (Regularity conditions)**  For each group $m$, assume $H_m := \mathbb{E}_{(x,y)\sim\mathcal{D}_m}\left[xx^T\right]$ (i.e., the data covariance matrix[1] of $\mathcal{D}_m$) and $\mathbb{E}_{(x,y)\sim\mathcal{D}_m}\left[y^2\right]$ exist and are finite. Furthermore, assume that $\text{tr}(H_m)$ is finite (i.e., $H_m$ is trace-class) and $H_m$ is symmetric positive definite (PSD). Let $\{\lambda_i(H_m)\}_{i=1}^{\infty}$ be the eigenvalues of $H_m$ sorted in non-increasing order.

Now, denote the population data covariance matrix $H := \mathbb{E}_{(x,y)\sim\mathcal{D}}\left[xx^T\right]$, and assume that $\text{tr}(H)$ is finite and $H$ is PSD. Let $\{\lambda_i(H)\}_{i=1}^{\infty}$ be the eigenvalues of $H$ sorted in non-increasing order. Furthermore, suppose the eigendecomposition of $H = \sum_i \lambda_i v_i v_i^T$; then, $H_{k:\infty} := \sum_{i>k} \lambda_i v_i v_i^T$. Similarly, the head of the identity matrix $I_{0:k} := \sum_{i=1}^{k} v_i v_i^T$. Additionally, let $\{\lambda_{(i)}(H)\}_{i=1}^{\infty}$ be the eigenvalues of $H$ such that $\left\{\frac{\left(1 - \left(1 - \gamma\lambda_{(i)}(H)\right)^N\right)^2}{\lambda_{(i)}(H)}\right\}_{i=1}^{\infty}$ is sorted in non-increasing order.

Similarly, let $\{\lambda_{[i]}(H)\}_{i=1}^{\infty}$ be the eigenvalues of $H$ such that $\left\{\frac{\left(1 - \left(1 - \gamma\lambda_{[i]}(H)\right)^N\right)^2}{\lambda_{[i]}^2(H)}\right\}_{i=1}^{\infty}$ is sorted in non-increasing order.

---

[1] We refer to $H_m$ as the data covariance matrix (following [1]), but it is in fact the raw second moment. We do **not** assume $\mathbb{E}_{(x,y)\sim\mathcal{D}_m}[x] = 0$.

We also define the following linear operators (which we assume to exist and be finite): $\mathcal{I} = I \otimes I$, $\mathcal{M} = \mathbb{E}_{(x,y)\sim\mathcal{D}}[x \otimes x \otimes x \otimes x]$ (where $\otimes$ is the tensor product), $\mathcal{T} = H \otimes I + I \otimes H - \gamma\mathcal{M}$. All the results about these operators from Lemma 4.1 in [1] hold.

**Assumption 2 (Fourth moment conditions)**    Assume that there exists a positive constant $\alpha > 0$, such that for any PSD matrix $A$, it holds that $\mathbb{E}_{(x,y)\sim\mathcal{D}}[xx^T A xx^T] \preceq \alpha \mathrm{tr}(HA) H$. This assumption is satisfied for Gaussian distributions by $\alpha = 3$, and it is further implied if the distribution over $H^{-\frac{1}{2}}x$ has sub-Gaussian tails (see Lemma A.1 in [1]).

**Assumption 3 (Noise conditions)**    Suppose that $\Sigma := \mathbb{E}_{(x,y)\sim\mathcal{D}}\left[(y - \langle w_m^*, x\rangle)^2 xx^T\right]$ (i.e., the covariance matrix of the gradient noise at $w_m^{*\,2}$) and $\sigma^2 := \left\|H^{-\frac{1}{2}}\Sigma H^{-\frac{1}{2}}\right\|_2$ exist and are finite. In effect, $\Sigma$ and $\sigma^2$ capture how poorly $w_m^*$ explains samples from the population distribution $\mathcal{D}$. If $w_m^*$ is optimal for $\mathcal{D}$, then $\sigma^2$ reduces to additive noise.

**Assumption 4 (Learning rate condition)**    Assume $\gamma < \frac{1}{\alpha\mathrm{tr}(H)}$.

The excess risk of a trained model for group $m$ quantifies how much worse the model performs for group $m$ than the optimal model for $m$ (i.e., the model parameterized by $w_m^*$) does. In Theorem 1, we present an excess risk bound of overparameterized linear regression with constant-stepsize SGD (with iterate averaging) for group $m$ in terms of the full eigenspectra of the data covariance matrices of the group and population.

Under the assumptions above, we are ready for the statement of the main theorem:

**Theorem 1**    We can bound the excess risk $\mathcal{E}_m$ for group $m$ as:

$$\mathcal{E}_m = \mathbb{E}_{\mathcal{D}}\left[L_{\mathcal{D}_m}(\bar{w}_N)\right] - L_{\mathcal{D}_m}(w_m^*) \leq 2 \cdot \mathrm{EffectiveBias} + 2 \cdot \mathrm{EffectiveVar},$$

where:

$$\mathrm{EffectiveBias} \leq \begin{cases} \frac{\lambda_1(H_m)\|w_0 - w_m^*\|_2^2}{\gamma^2 N^2 \lambda_{[1]}^2(H)}, & \lambda_{[1]}(H) \geq \frac{1}{\gamma N} \\ \lambda_1(H_m)\|w_0 - w_m^*\|_2^2, & \text{otherwise} \end{cases}$$

$$\mathrm{EffectiveVar} \leq \frac{\frac{2\alpha}{N\gamma}\left(\|w_0 - w_m^*\|_{I_{0:k^*}}^2 + N\gamma\|w_0 - w_m^*\|_{H_{k^*:\infty}}^2\right) + \sigma^2}{1 - \gamma\alpha\mathrm{tr}(H)}$$

$$\cdot \left( \frac{1}{N}\underbrace{\sum_{i\in\delta}\frac{\lambda_i(H_m)}{\lambda_{(i)}(H)}}_{\text{head}} + N\gamma^2 \underbrace{\sum_{i\notin\delta}\lambda_i(H_m)\lambda_{(i)}(H)}_{\text{tail}}\right),$$

where $k^* := \max\left\{k : \lambda_k(H) \geq \frac{1}{\gamma N}\right\}$ and $\delta := \left\{k : \lambda_{(k)}(H) \geq \frac{1}{\gamma N}\right\}$. It is easy to show that $\delta$ is finite and $\forall i, |(i) - i| \leq k^*$. Furthermore, as the bound suggests, to obtain a vanishing bound, we need:

1. A sufficiently large sample from the population: $\lambda_{[1]}(H) \geq \frac{1}{\gamma N}$.

2. The head to converge in $N$: $\sum_{i\in\delta}\frac{\lambda_i(H_m)}{\lambda_{(i)}(H)} = o(N)$.

3. The tail to converge in $N$: $\sum_{i\notin\delta}\lambda_i(H_m)\lambda_{(i)}(H) = o(1/N)$.

We prove Theorem 1 in Section A. In the following section, we contextualize the group excess risk bound through the lens of real-world challenges in practicing trustworthy machine learning.

---

[2]Recall that $w_m^*$ are the optimal parameters for group $m$.

### 4.1 Interpreting the group excess risk bound

In interpreting the bound for $\mathcal{E}_m$, we consider the case where the eigenspectrum of $H$ rapidly decays and thus focus on the head of the bound.[3] We note that the head does not necessarily have a weaker convergence criterion, as although the head is a finite sum, $k^*$ depends on $N$. Re-expressing the head of the bound:

$$\sum_{i \in \delta} \frac{\lambda_i(H_m)}{\lambda_{(i)}(H)} = 2 \underbrace{\sum_{i \in \delta} \mathrm{KL}\,(p_i || q_i)}_{\text{distributional difference}} + \underbrace{\left[ \sum_{i \in \delta} \log \lambda_i(H_m) - \sum_{i \in \delta} \log \lambda_{(i)}(H) \right]}_{\text{relative feature variance}} + \underbrace{|\delta|}_{\text{group effective dimension}},$$

where $p_i = \mathcal{N}\left(0, \lambda_i(H_m)\right)$ and $q_i = \mathcal{N}\left(0, \lambda_{(i)}(H)\right)$ [29]. This result shows that the excess risk for group $m$ can be minimized (and thus generalization to group $m$ can be improved) by:

1. *Making the feature distributional difference between group $m$ and the population smaller.* This result corroborates findings in the fairness literature that *randomly* oversampling or increasing training data from minoritized groups (thereby boosting the representation of group $m$ in the population) may improve group generalization [19].

2. *Minimizing the variance of feature values in group $m$ relative to the variance of feature values in the population.* High relative feature variance can occur when group $m$ has sparse or noisy data, which poses a challenge in the real world because minoritized groups are often sidelined in data collection [30] and their data may only be partially observed [31]. This finding is also consistent with the literature on SGD's implicit bias to rely less on high-variance features [8].

3. *Reducing the group effective dimension of SGD.* Recall that $|\delta| = \left| \left\{ k : \lambda_{(k)}(H) \geq \frac{1}{\gamma N} \right\} \right|$; therefore, $|\delta|$ can be reduced by: 1) reducing the learning rate $\gamma$, and 2) reducing the number of training samples $N$.

While, theoretically, it seems that increasing the representation of minoritized groups in the training data and better including them in data collection improves generalization to such groups, it is important to not engage in predatory inclusion and exploitative data collection practices[4]. We further emphasize that simply increasing the sheer number of samples, especially without analyzing the *randomness* or validity of sampling strategies, does not imply increasing the representation of minoritized groups in the training data [32]. Overall, we believe that one of the first steps of socially conscientious data work is to consider how data collection practices reinforce and contribute to the power relations and complex social inequality experienced by minoritized groups [33].

## 5 Empirical results

To investigate our group excess risk bound, we empirically examine how well our bound aligns with the real group excess risk in a simulated setting, wherein we have control over $\mathcal{D}$, $\mathcal{D}_m$, and $w_m^*$. In particular, we assume $\mathcal{D} := p\mathcal{D}_m + (1-p)\mathcal{D}_{\text{rest}}$ is a mixture distribution for $p \in [0, 1]$. If $p << 0.5$, $m$ could be considered a minoritized group, and the excess risk for group $m$ would have implications for fairness (Section 1). If $p >> 0.5$, $\mathcal{D}_{\text{rest}}$ could be viewed as noise, so a model's excess risk for group $m$ would offer insight into the robustness of the model (Section 1). $p = 0$ models a privacy-risk or OOD setting, as $\mathcal{D}_m$ would be an extra-population group (Section 1). In our experiments, we compare the group excess risk and our bound thereof for various values of $p \in [0, 1]$. Our code may be found at: `https://github.com/ArjunSubramonian/group-excess-risk-bound-sgd.git`.

We mostly use the same experimental hyperparameters as [1]: $N = 200$, $d = 2000 >> N$ (to simulate overparameterization), $\alpha = 3$, and $\gamma = \frac{0.99}{\alpha tr(H)}$. We assume $\mathcal{D}_m$ and $\mathcal{D}_{\text{rest}}$ are *well-specified*. That is, we generate $(x_m, y_m) \sim \mathcal{D}_m$ as $x_m \sim \mathcal{N}(\mu_m, \Sigma_m)$ and $y_m = \langle w_m^*, x_m \rangle + \epsilon_m$ (where $\epsilon_m \sim \mathcal{N}(0, 1)$). Similarly, we generate $(x_{\text{rest}}, y_{\text{rest}}) \sim \mathcal{D}_{\text{rest}}$ as $x_{\text{rest}} \sim \mathcal{N}(\mu_{\text{rest}}, \Sigma_{\text{rest}})$ and $y_m = \langle w_{\text{rest}}^*, x_{\text{rest}} \rangle$ (where $w_{\text{rest}}^*$ are the optimal parameters for $\mathcal{D}_{\text{rest}}$). Inspired by [1] (Section

---

[3]We leave rigorously analyzing the tail of the bound as future work. If the eigenspectrum of $H$ doesn't decay rapidly (i.e., there exist many high-variance features), then the variance error of the group excess risk will be higher.

[4]`https://slideslive.com/38955136/beyond-the-fairness-rhetoric-in-ml`

6), we consider two overparameterized linear regression settings (1 and 2) with different rates of eigenspectrum decay for $H$ and $H_m$ that satisfy our assumptions from Section 4:

1. $\mu_{\text{rest}}[i] := 0, \mu_m[i] := 0, \Sigma_{\text{rest}} := \text{diag}\left(\{(i+1)^{-1}\log(i+1)^{-2}\}_{i=1}^d\right), \Sigma_m := \beta_1 \times \Sigma_{\text{rest}}, N := \beta_2 \times 200$

2. $\mu_{\text{rest}}[i] := 0, \mu_m[i] := 0, \Sigma_{\text{rest}} := \text{diag}\left(\{i^{-2}\}_{i=1}^d\right), \Sigma_m := \beta_1 \times \Sigma_{\text{rest}}, N := \beta_2 \times 200,$

where $\beta_1 > 0$ affects the *distributional difference* and *relative feature variance* and a smaller $\beta_2 > 0$ affects $\delta$. Recall that $H = pH_m + (1-p)H_{\text{rest}}$, with $H_m = \Sigma_m + \mu_m\mu_m^T$ and $H_{\text{rest}} = \Sigma_{\text{rest}} + \mu_{\text{rest}}\mu_{\text{rest}}^T = \Sigma_{\text{rest}}$. For both settings 1 and 2, $w_m^*[i] = i^{-1}$ and $w_{\text{rest}}^*[i] = 2 \times i^{-1}$. $w_0$ is set using the Kaiming Uniform method[5]. We choose sufficiently different $w_m^*, w_{\text{rest}}^*$ to enlarge the total variation distance between $\mathcal{D}_m$ and $\mathcal{D}_{\text{rest}}$, towards stress-testing our group excess risk bound.

To compute our group excess risk bound, we need to upper bound $\sigma^2$. $\Sigma = p\mathbb{E}_{\mathcal{D}_m}\left[(y - \langle w_m^*, x\rangle)^2 xx^T\right] + (1-p)\mathbb{E}_{\mathcal{D}_{\text{rest}}}\left[(y - \langle w_m^*, x\rangle)^2 xx^T\right] = pH_m + (1-p)\mathbb{E}_{\mathcal{D}_{\text{rest}}}\left[(\langle w_m^*, x\rangle)^2 xx^T\right] = pH_m + (1-p)\mathbb{E}_{\mathcal{D}_{\text{rest}}}\left[xx^T\left(w_m^*w_m^{*T}\right)xx^T\right]$. By Assumption 2, $\Sigma \preceq pH_m + (1-p)\alpha\text{tr}\left(H_{\text{rest}}w_m^*w_m^{*T}\right)H_{\text{rest}}$. Hence, $\sigma^2 \leq \left\|H^{-\frac{1}{2}}\left(pH_m + (1-p)\alpha w_m^{*T}H_{\text{rest}}w_m^*H_{\text{rest}}\right)H^{-\frac{1}{2}}\right\|_2$.

Our empirical results are displayed in Figure 2; please note that there is a dual y-axis. In all the plots, the group excess risk decreases (i.e., generalization improves) when $p$ increases, as $\mathcal{D}_m$ is (randomly) sampled at a higher rate during training. This finding corroborates our commentary about fairness, privacy, robustness, and OOD generalization. Furthermore, the plots demonstrate that our group excess risk bound closely captures the trend of the true excess risk as $p$ increases from 0 to 1. Now, we contextualize our experiments in terms of our interpretation of the group excess risk bound from Section 4.1:

- **$\beta_1$** : Holding $\beta_2$ constant, reducing $\beta_1$ from 1 to 0.5 (which increases the *distributional difference*) appears to lower (rather than increase) the true excess risk and bound. This is because reducing $\beta_1$ also reduces the *relative feature variance*.

- **$\beta_2$** : Holding $\beta_1$ constant, in setting 2, reducing $\beta_2$ from 1 to 0.5 (which shrinks the cardinality of $\delta$) seems to lower the true excess risk and bound. This can be attributed to fewer terms in the summation and a smaller SGD *group effective dimension*. However, in setting 1, reducing $\beta_2$ from 1 to 0.5 does not affect the true excess risk or lower bound, likely since in setting 1, the cardinality of $\delta$ is not reduced as substantially as it is in setting 2.

## 6 Discussion and conclusion

In this paper, we characterize the inherent generalization of overparameterized linear regression with constant-stepsize SGD (with iterate averaging) to groups within and outside the population from which training instances are sampled. We do so by proving the excess risk bound for an arbitrary group in terms of the full eigenspectra of the data covariance matrices of the group and population. We additionally present a novel interpretation of the group excess risk bound through the lens of real-world challenges in practicing trustworthy machine learning. Finally, we empirically study our bound on simulated data.

This paper offers numerous promising future directions for research. We encourage proving a lower bound on the group excess risk to determine if our upper bound is indeed tight (up to constant factors). We also suggest proving group excess risk bounds for tail averaging and last-iterate SGD with learning rate decay [1, 2], as well as minibatch GD. It would further be interesting to extend this work to prove group excess risk bounds for logistic regression, 2-layer neural networks, and 1-layer graph convolutional networks [34].

---

[5]https://pytorch.org/docs/stable/generated/torch.nn.Linear.html

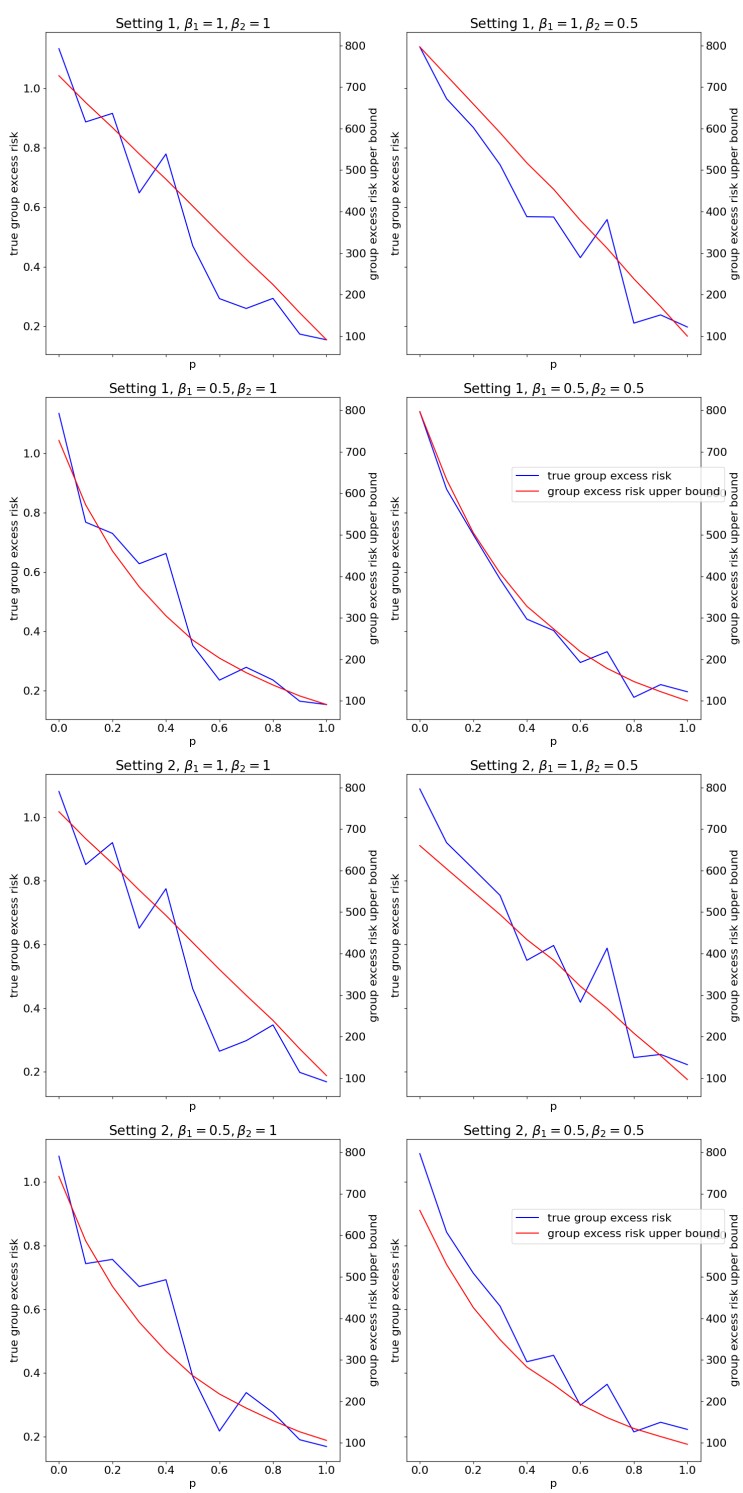

Figure 2: True group excess risk and our bound thereof for $p \in \{0.0, 0.1, 0.2, \ldots, 0.9, 1.0\}$ for settings 1 and 2. Each data point in the plots is averaged over 10 independent runs, and the true group excess risk data points are approximated over $10^5$ samples from $\mathcal{D}_m$.

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

# A Proof of the main result

## A.1 Bias-variance decomposition

Towards proving our main result, we first decompose the excess risk for group $m$ into bias and variance errors. We define the centered SGD iterate $\eta_t := w_t - w_m^*$; similarly, $\bar{\eta}_N := \frac{1}{N} \sum_{t=0}^{N-1} \eta_t$. [1] (Equation 4.2) shows the bias-variance decomposition of the iterate:

$$\eta_t = \eta_t^{\text{bias}} + \eta_t^{\text{var}}$$
$$\eta_t^{\text{bias}} = \left(I - \gamma x_t x_t^T\right) \eta_{t-1}^{\text{bias}}, \eta_0^{\text{bias}} = \eta_0$$
$$\eta_t^{\text{var}} = \left(I - \gamma x_t x_t^T\right) \eta_{t-1}^{\text{var}} + \gamma \xi_t x_t, \eta_0^{\text{var}} = 0,$$

where $\xi_t := y_t - \langle w_m^*, x_t \rangle$ is the inherent noise. [1] (Equations 4.4 and 4.5) then proves the recursive forms:

$$B_t := \mathbb{E}_{(x,y)\sim\mathcal{D}} \left[\eta_t^{\text{bias}} \otimes \eta_t^{\text{bias}}\right] = (\mathcal{I} - \gamma\mathcal{T}) \circ B_{t-1}, B_0 = \eta_0 \otimes \eta_0$$
$$C_t := \mathbb{E}_{(x,y)\sim\mathcal{D}} \left[\eta_t^{\text{var}} \otimes \eta_t^{\text{var}}\right] = (\mathcal{I} - \gamma\mathcal{T}) \circ C_{t-1} + \gamma^2\Sigma, C_0 = 0,$$

where $\mathcal{O} \circ A$ denotes an operator $\mathcal{O}$ applied to a symmetric matrix $A$. We also define $\bar{\eta}_N^{\text{bias}} := \frac{1}{N} \sum_{t=0}^{N-1} \eta_t^{\text{bias}}$ and $\bar{\eta}_N^{\text{var}} := \frac{1}{N} \sum_{t=0}^{N-1} \eta_t^{\text{var}}$, and see that $\bar{\eta}_t = \bar{\eta}_t^{\text{bias}} + \bar{\eta}_t^{\text{var}}$.

Without loss of generality, we present a bias-variance decomposition of the excess risk for group $m$:

$$\mathcal{E}_m = \mathbb{E}_{\mathcal{D}} \left[L_{\mathcal{D}_m} (\bar{w}_N) - L_{\mathcal{D}_m} (w_m^*)\right]$$
$$= \mathbb{E}_{\mathcal{D}} \left[\frac{1}{2}\mathbb{E}_{(x,y)\sim\mathcal{D}_m} \left(y - \bar{w}_N^T x\right)^2 - \frac{1}{2}\mathbb{E}_{(x,y)\sim\mathcal{D}_m} \left(y - {w_m^*}^T x\right)^2\right]$$
$$= \frac{1}{2}\mathbb{E}_{\mathcal{D}} \left[\mathbb{E}_{(x,y)\sim\mathcal{D}_m} \left((w_m^* - \bar{w}_N)^T x + \xi\right)^2 - \mathbb{E}_{(x,y)\sim\mathcal{D}_m}\xi^2\right]$$
$$= \frac{1}{2}\mathbb{E}_{\mathcal{D}} \left[\mathbb{E}_{(x,y)\sim\mathcal{D}_m} \left[(\bar{w}_N - w_m^*)^T xx^T (\bar{w}_N - w_m^*) - 2(\bar{w}_N - w_m^*)^T (\xi x)\right]\right]$$
$$= \frac{1}{2}\mathbb{E}_{\mathcal{D}} \left[(\bar{w}_N - w_m^*)^T \mathbb{E}_{(x,y)\sim\mathcal{D}_m} \left[xx^T\right] (\bar{w}_N - w_m^*)\right]$$
$$= \frac{1}{2} \langle H_m, \mathbb{E}_{\mathcal{D}} [\bar{\eta}_N \otimes \bar{\eta}_N]\rangle$$

Then, by Lemma B.2 from [1] and Young's inequality, $\mathbb{E}_{\mathcal{D}} \left[L_{\mathcal{D}_m} (\bar{w}_N) - L_{\mathcal{D}_m} (w_m^*)\right] \leq \left(\sqrt{\text{bias}} + \sqrt{\text{var}}\right)^2 \leq 2 \cdot \text{bias} + 2 \cdot \text{var}$, where bias $:= \frac{1}{2} \langle H_m, \mathbb{E}_{\mathcal{D}} \left[\bar{\eta}_N^{\text{bias}} \otimes \bar{\eta}_N^{\text{bias}}\right]\rangle$ and var $:= \frac{1}{2} \langle H_m, \mathbb{E}_{\mathcal{D}} [\bar{\eta}_N^{\text{var}} \otimes \bar{\eta}_N^{\text{var}}]\rangle$. The bias and var errors provide a decomposition of a bound for $\mathcal{E}_m$.

## A.2 Bounding the bias error

Now that we have decomposed a bound for $\mathcal{E}_m$ into bias and variance errors, we will separately bound these errors in terms of the full eigenspectra of the data covariance matrices of group $m$ and the population, and then combine these bounds to achieve the desired bound for $\mathcal{E}_m$. In Theorem 2, we focus on the bias error.

**Theorem 2** We can bound the bias error as:

$$\text{bias} \leq \frac{\alpha\text{tr}(B_{0,N})}{N\gamma(1 - \gamma\alpha\text{tr}(H))} \left(\frac{1}{N}\sum_{i\in\delta} \frac{\lambda_i(H_m)}{\lambda_{(i)}(H)} + N\gamma^2\sum_{i\notin\delta}\lambda_i(H_m)\lambda_{(i)}(H)\right)$$
$$+ \begin{cases} \frac{\lambda_1(H_m)\|w_0 - w_m^*\|_2^2}{\gamma^2 N^2\lambda_{[1]}^2(H)}, & \lambda_{[1]}(H) \geq \frac{1}{\gamma N} \\ \lambda_1(H_m)\|w_0 - w_m^*\|_2^2, & \text{otherwise} \end{cases},$$

where $\text{tr}(B_{0,N}) \leq 2\left(\|w_0 - w_m^*\|_{I_{0:k^*}}^2 + N\gamma\|w_0 - w_m^*\|_{H_{k^*:\infty}}^2\right)$ (where $k^* :=$ $\max\left\{k : \lambda_k(H) \geq \frac{1}{\gamma N}\right\}$) and $\delta := \left\{k : \lambda_{(k)}(H) \geq \frac{1}{\gamma N}\right\}$.

**Proof of Theorem 2** [1] (Lemma B.3) shows that:

$$\mathbb{E}_{\mathcal{D}}\left[\bar{\eta}_N^{\text{bias}} \otimes \bar{\eta}_N^{\text{bias}}\right] \preceq \frac{1}{N^2} \sum_{t=0}^{N-1} \sum_{k=t}^{N-1} (I - \gamma H)^{k-t} \mathbb{E}_{\mathcal{D}}\left[\eta_t^{\text{bias}} \otimes \eta_t^{\text{bias}}\right] + \mathbb{E}_{\mathcal{D}}\left[\eta_t^{\text{bias}} \otimes \eta_t^{\text{bias}}\right] (I - \gamma H)^{k-t}$$

Because $H$ and $H_m$ are PSD, and $H^{-1}$ commutes with $(I - \gamma H)^{k-t}$:

$$\begin{aligned}
\text{bias} &= \frac{1}{2} \left\langle H_m, \mathbb{E}_{\mathcal{D}}\left[\bar{\eta}_N^{\text{bias}} \otimes \bar{\eta}_N^{\text{bias}}\right] \right\rangle \\
&\leq \frac{1}{2N^2} \sum_{t=0}^{N-1} \sum_{k=t}^{N-1} \left\langle H_m, (I - \gamma H)^{k-t} \mathbb{E}_{\mathcal{D}}\left[\eta_t^{\text{bias}} \otimes \eta_t^{\text{bias}}\right] + \mathbb{E}_{\mathcal{D}}\left[\eta_t^{\text{bias}} \otimes \eta_t^{\text{bias}}\right] (I - \gamma H)^{k-t} \right\rangle \\
&= \frac{1}{2N^2} \sum_{t=0}^{N-1} \sum_{k=t}^{N-1} \left\langle (I - \gamma H)^{k-t} H_m + H_m (I - \gamma H)^{k-t}, \mathbb{E}_{\mathcal{D}}\left[\eta_t^{\text{bias}} \otimes \eta_t^{\text{bias}}\right] \right\rangle \\
&= \frac{1}{2\gamma N^2} \sum_{t=0}^{N-1} \left\langle H_m H^{-1}\left(I - (I - \gamma H)^{N-t}\right) + \left(I - (I - \gamma H)^{N-t}\right) H^{-1} H_m, B_t \right\rangle \\
&\leq \frac{1}{2\gamma N^2} \left\langle H_m H^{-1}\left(I - (I - \gamma H)^{N}\right) + \left(I - (I - \gamma H)^{N}\right) H^{-1} H_m, \sum_{t=0}^{N-1} B_t \right\rangle
\end{aligned}$$

[1] (Lemma B.10) shows that:

$$\sum_{t=0}^{N-1} B_t \preceq \sum_{k=0}^{N-1} (I - \gamma H)^k \left( \frac{\gamma \alpha \text{tr}(B_{0,N})}{1 - \gamma \alpha \text{tr}(H)} \cdot H + B_0 \right) (I - \gamma H)^k,$$

where $B_{0,N} = B_0 - (I - \gamma H)^N B_0 (I - \gamma H)^N$. Furthermore, [1] (Lemma B.11) shows that $\text{tr}(B_{0,N}) \leq 2\left(\|w_0 - w_m^*\|_{I_{0:k^*}}^2 + N\gamma \|w_0 - w_m^*\|_{H_{k^*:\infty}}^2\right)$, where $k^* := \max\left\{k : \lambda_{(k)}(H) \geq \frac{1}{\gamma N}\right\}$. Therefore:

$$\begin{aligned}
\text{bias} &\leq \frac{1}{2\gamma N^2} \sum_{k=0}^{N-1} \left\langle H_m H^{-1}\left(I - (I - \gamma H)^{N}\right) + \left(I - (I - \gamma H)^{N}\right) H^{-1} H_m, \right. \\
&\qquad\qquad \left. (I - \gamma H)^k \left( \frac{\gamma \alpha \text{tr}(B_{0,N})}{1 - \gamma \alpha \text{tr}(H)} \cdot H + B_0 \right) (I - \gamma H)^k \right\rangle \\
&\leq \frac{1}{2\gamma N^2} \sum_{k=0}^{N-1} \left\langle H_m H^{-1}\left(I - (I - \gamma H)^{N}\right)(I - \gamma H)^k, \frac{\gamma \alpha \text{tr}(B_{0,N})}{1 - \gamma \alpha \text{tr}(H)} \cdot H + B_0 \right\rangle \\
&\qquad\qquad + \left\langle (I - \gamma H)^k \left(I - (I - \gamma H)^{N}\right) H^{-1} H_m, \frac{\gamma \alpha \text{tr}(B_{0,N})}{1 - \gamma \alpha \text{tr}(H)} \cdot H + B_0 \right\rangle \\
&= \frac{1}{2\gamma N^2} \sum_{k=0}^{N-1} \left\langle (I - \gamma H)^k - (I - \gamma H)^{N+k}, \frac{\gamma \alpha \text{tr}(B_{0,N})}{1 - \gamma \alpha \text{tr}(H)} \cdot H H_m H^{-1} + B_0 H_m H^{-1} \right. \\
&\qquad\qquad\qquad\qquad \left. + H^{-1} H_m H \cdot \frac{\gamma \alpha \text{tr}(B_{0,N})}{1 - \gamma \alpha \text{tr}(H)} + H^{-1} H_m B_0 \right\rangle,
\end{aligned}$$

where we use that $(I - \gamma H)^k \preceq I$. We now define the following terms:

$$I_1 = \frac{\alpha \text{tr}(B_{0,N})}{2N^2 (1 - \gamma \alpha \text{tr}(H))} \sum_{k=0}^{N-1} \left\langle (I - \gamma H)^k - (I - \gamma H)^{N+k}, H H_m H^{-1} \right\rangle$$

$$I_2 = \frac{1}{2\gamma N^2} \sum_{k=0}^{N-1} \left\langle (I - \gamma H)^k - (I - \gamma H)^{N+k}, B_0 H_m H^{-1} \right\rangle$$

We first focus on bounding $I_1$. By von Neumann's trace inequality, if two matrices $A$ and $B$ are PSD, then $\text{tr}(AB) \leq \sum_i \lambda_i(A)\lambda_i(B)$, where $\{\lambda_i(A)\}_{i=1}^{\infty}$ and $\{\lambda_i(B)\}_{i=1}^{\infty}$ are the eigenvalues of $A$ and $B$ in non-increasing order, respectively [35]. Thus, leveraging the joint diagonalizability of $H$ and $I - \gamma H$:

$$
\begin{aligned}
I_1 &= \frac{\alpha \text{tr}(B_{0,N})}{2N^2(1 - \gamma\alpha\text{tr}(H))} \sum_{k=0}^{N-1} \left\langle (I - \gamma H)^k \left( I - (I - \gamma H)^N \right), H H_m H^{-1} \right\rangle \\
&= \frac{\alpha \text{tr}(B_{0,N})}{2\gamma N^2(1 - \gamma\alpha\text{tr}(H))} \left\langle \left( I - (I - \gamma H)^N \right)^2 H^{-1}, H H_m H^{-1} \right\rangle \\
&= \frac{\alpha \text{tr}(B_{0,N})}{2\gamma N^2(1 - \gamma\alpha\text{tr}(H))} \left\langle \left( I - (I - \gamma H)^N \right)^2 H^{-1}, H_m \right\rangle \\
&\leq \frac{\alpha \text{tr}(B_{0,N})}{2\gamma N^2(1 - \gamma\alpha\text{tr}(H))} \sum_i \frac{\left( 1 - \left( 1 - \gamma\lambda_{(i)}(H) \right)^N \right)^2 \lambda_i(H_m)}{\lambda_{(i)}(H)} \\
&\leq \frac{\alpha \text{tr}(B_{0,N})}{2\gamma N^2(1 - \gamma\alpha\text{tr}(H))} \sum_i \frac{\min\left\{ 1, \gamma^2 N^2 \lambda_{(i)}^2(H) \right\} \cdot \lambda_i(H_m)}{\lambda_{(i)}(H)} \\
&= \frac{\alpha \text{tr}(B_{0,N})}{2N\gamma(1 - \gamma\alpha\text{tr}(H))} \left( \frac{1}{N} \sum_{i \in \delta} \frac{\lambda_i(H_m)}{\lambda_{(i)}(H)} + N\gamma^2 \sum_{i \notin \delta} \lambda_i(H_m)\lambda_{(i)}(H) \right),
\end{aligned}
$$

where $\delta := \left\{ k : \lambda_{(k)}(H) \geq \frac{1}{\gamma N} \right\}$. Because $\text{tr}(H)$ is finite, $\sum_{i \notin \delta} \lambda_{(i)}(H)$ converges. Similarly, $\text{tr}(H_m)$ is finite, so $\sum_{i \notin \delta} \lambda_i(H_m)$ converges. Therefore, by Abel's Lemma, $\sum_{i \notin \delta} \lambda_i(H_m)\lambda_{(i)}(H)$ converges. It is easy to show that $\sum_{i \notin \delta} \lambda_i(H_m)\lambda_{(i)}(H)$ converges similarly to $\sum_{i \notin \delta} \lambda_i(H_m)\lambda_i(H)$ because $\forall i, |(i) - i| \leq k^*$.

Now, we bound $I_2$. Because $B_0 = (w_0 - w_m^*)(w_0 - w_m^*)^T$, the largest and only non-zero eigenvalue of $B_0$ is $\lambda_1(B_0) = \|w_0 - w_m^*\|_2^2$. Therefore:

$$
\begin{aligned}
I_2 &= \frac{1}{2\gamma^2 N^2} \left\langle \left( I - (I - \gamma H)^N \right)^2 H^{-2}, B_0 H_m \right\rangle \\
&\leq \frac{1}{2\gamma^2 N^2} \cdot \frac{\lambda_1(H_m)\|w_0 - w_m^*\|_2^2}{\lambda_{[1]}^2(H)} \cdot \begin{cases} 1, & \lambda_{[1]}(H) \geq \frac{1}{\gamma N} \\ \gamma^2 N^2 \lambda_{[1]}^2(H), & \text{otherwise} \end{cases} \\
&= \begin{cases} \frac{\lambda_1(H_m)\|w_0 - w_m^*\|_2^2}{2\gamma^2 N^2 \lambda_{[1]}^2(H)}, & \lambda_{[1]}(H) \geq \frac{1}{\gamma N} \\ \frac{1}{2}\lambda_1(H_m)\|w_0 - w_m^*\|_2^2, & \text{otherwise} \end{cases}.
\end{aligned}
$$

Now that we have bounds for $I_1$ and $I_2$, leveraging the joint diagonalizability of $H$ and $I - \gamma H$:

$$
\begin{aligned}
\text{bias} \leq {} & \frac{\alpha \text{tr}(B_{0,N})}{N\gamma(1 - \gamma\alpha\text{tr}(H))} \left( \frac{1}{N} \sum_{i \in \delta} \frac{\lambda_i(H_m)}{\lambda_{(i)}(H)} + N\gamma^2 \sum_{i \notin \delta} \lambda_i(H_m)\lambda_{(i)}(H) \right) \\
& + \begin{cases} \frac{\lambda_1(H_m)\|w_0 - w_m^*\|_2^2}{\gamma^2 N^2 \lambda_{[1]}^2(H)}, & \lambda_{[1]}(H) \geq \frac{1}{\gamma N} \\ \lambda_1(H_m)\|w_0 - w_m^*\|_2^2, & \text{otherwise} \end{cases}.
\end{aligned}
$$

Thus, we have successfully bounded the bias error in terms of the full eigenspectra of the data covariance matrices of group $m$ and the population.

### A.3 Bounding the variance error

Proceeding in our journey to prove Theorem 1, we will bound the variance error.

**Theorem 3** We can bound the variance error as:

$$\text{var} \leq \frac{\sigma^2}{1 - \gamma\alpha\text{tr}\,(H)} \left( \frac{1}{N} \sum_{i \in \delta} \frac{\lambda_i \, (H_m)}{\lambda_{(i)} \, (H)} + N\gamma^2 \sum_{i \notin \delta} \lambda_i \, (H_m) \, \lambda_{(i)} \, (H) \right)$$

Similar to for the bias error:

$$\text{var} \leq \frac{1}{2\gamma N^2} \sum_{t=0}^{N-1} \left\langle H_m H^{-1} \left( I - (I - \gamma H)^{N-t} \right) + \left( I - (I - \gamma H)^{N-t} \right) H^{-1} H_m, C_t \right\rangle$$

[1] (Lemma B.5) shows $C_t \preceq \frac{\gamma\sigma^2}{1-\gamma\alpha\text{tr}(H)} \left( I - (I - \gamma H)^t \right)$. Therefore, similar to for the bias error:

$$\text{var} \leq \frac{\sigma^2}{2N^2 \, (1 - \gamma\alpha\text{tr}\,(H))} \sum_{t=0}^{N-1} \left\langle H_m H^{-1} \left( I - (I - \gamma H)^{N-t} \right) + \left( I - (I - \gamma H)^{N-t} \right) H^{-1} H_m, \right.$$

$$\left. I - (I - \gamma H)^t \right\rangle$$

$$= \frac{\sigma^2}{2N^2 \, (1 - \gamma\alpha\text{tr}\,(H))} \left\langle \sum_{t=0}^{N-1} \left( I - (I - \gamma H)^{N-t} \right) \left( I - (I - \gamma H)^t \right), H_m H^{-1} + H^{-1} H_m \right\rangle$$

$$\leq \frac{\sigma^2}{2N \, (1 - \gamma\alpha\text{tr}\,(H))} \left\langle \left( I - (I - \gamma H)^N \right)^2, H_m H^{-1} + H^{-1} H_m \right\rangle$$

$$\leq \frac{\sigma^2}{1 - \gamma\alpha\text{tr}\,(H)} \left( \frac{1}{N} \sum_{i \in \delta} \frac{\lambda_i \, (H_m)}{\lambda_{(i)} \, (H)} + N\gamma^2 \sum_{i \notin \delta} \lambda_i \, (H_m) \, \lambda_{(i)} \, (H) \right)$$

Having appropriately bounded both the bias and variance errors, we have proved Theorem 1.

