# OpenReview forum: "Group Excess Risk Bound of Overparameterized Linear Regression with Constant-Stepsize SGD"
_NeurIPS.cc/2022/Workshop/TSRML — TSRML2022_

### Official Review · Reviewer_v1W4 · 2022-10-17

**Overall Rating:** 5

**Summary:**

The authors consider the setting of over-parameterized linear regression (via infinite feature dimensionality) to derive a worst-case bound on an arbitrary group's excess risk (difference in MSE loss) of the expected parameterization for a specific SGD training and the optimal parameterization for this group. Under some conditions on the step size, parameter averaging, and considered data distributions, they obtain an interpretable bound based on the eigenspectra of the considered data distributions and the effective dimensionality of the SGD optimization.

**Strengths:**

* The presented insights are applicable to OOD generalization, fairness, and robustness, thus providing interesting insights into many important topics.
* The problem setup and the limitations/conditions are well explained
* Intuitive explanation of the conditions to be satisfied to have small excess risk bound.


**Weaknesses:**

* Strong limitations of the setting (linear regression, infinite feature dimensionality, specific SGD optimization mode, constraints on feature distributions) seem not sufficiently discuss and lead to questions regarding the applicability of the obtained insights.
* The interpretation focuses on the head of the distribution, which has much weaker convergence critertion (finite sum over ratio of eigenvalues has to be O(N)) for the bound to vanish and might be less relevant in a setting where minority groups/outliers are of great importance.
* Empirical evaluation only for Gaussian distribution with identical eigenspectra. Further choice of $\gamma = 1/(\alpha tr(H))$  seems like it should let the bound on the effective variance go to infinity via division by 0 (The learning rate condition might have to be strict < instead of $\leq$).
* The claim of bound tightness being confirmed by two experiments on mixtures of Gaussians with identical eigenspectra seems questionable.
* The insights obtained from the analysis in Section 4.1 seem to be partially non-actionable (e.g. reducing the distributional differences via oversampling is only feasible for very few minority groups and reducing feature variance is only possible if variance is mostly noise induced), partially contradicting (e.g. the variance of the overall distribution is increased if we oversample a minority distribution with different mode).

**Overall Recommendation:**

The paper tackles an interesting problem and has the potential to provide interesting insights into multiple important issues around trustworthy ML. However, in its current state, the motivation for many assumptions is not discussed and the lack of a proof sketch in the main paper makes their significance hard to judge. While the derived insights on how to improve the excess risk bounds are interesting, they rely on even stronger assumptions and seem largely non-actionable (e.g. consider more similar distributions) or non-surprising (e.g. consider more samples). Finally, the empirical evaluation provides interesting insights but considers only a special case where many parts of the bound become trivial, making it unsuitable to judge its tightness. Further, while two settings are considered the observed effects are not discussed. I would thus like to encourage the authors to provide more intuitions regarding the derivation of Theorem 1 and a more in-depth analysis of both their theoretical and empirical insights.

**Review Confidence:**

3: The reviewer is fairly confident that the evaluation is correct

---

### Official Review · Reviewer_n3JH · 2022-10-20
**Understanding generalization under overparameterized linear model with fixed step sized SGD**

**Overall Rating:** 5

**Summary:**

This paper develops theories on the excess risk bound $\varepsilon \le \text{Bias} + \text{Var}$ in terms of eigenspectra of the data covariance matrices of the group and population when working with the simple case of overparameterized linear regression SGD with a constant step size.
The authors present scenarios in which the resulting theory can be related to in terms of fairness and trustworthy machine learning, for instance, how minority groups are affected for generalization depending on sampling strategy, feature variance, or data collection practice.
They further design an experiment to show the group excess risk decreases as the group becomes larger.

**Strengths:**

- present realistic scenarios related to fairness
- translate theoretical results into words and show how these findings are related to others in the literature
- conduct experiments to achieve empirical evidence -- excess risk decreasing as with increasing p and seemingly tight -- to support the theory and relate it to the concepts of fairness, privacy, and out of distribution generalization


**Weaknesses:**

- theory developed for a extremely simplified model, i.e. overparameterized linear model with unrealistic setting where $x$ and $w$ go to infinity, remaining as a quite theoretical work
- insufficient experiments and lack of analysis to support how it will generalize to more "real" models
- often many parts unclearly written making it hard to interpret or give feedback


**Overall Recommendation:**

While it is appreciated the theoretical contribution for characterizing the group excessive risk as a tool to understand generalization to intra and ood groups, the results do not seem to add much significant insights beyond what is known. The experiments are designed rather naively and satisfy quite obvious expectations.

**Review Confidence:**

3: The reviewer is fairly confident that the evaluation is correct

---

### Decision · Program_Chairs · 2022-10-23

**Decision:**

Accept

**Comment:**

The paper derived a excess risk bound for SGD-learned overparameterized linear regression. The results can be useful when dealing with group-imbalanced data. Reviewers generally found that some assumptions in this paper are too strong and not applicable to real settings, however I feel it is acceptable as a theory paper. The topic is relevant to the workshop and the topic understudy is important so I accept the paper to allow more communications among the authors and other experts in this field.